# Tissue-resident memory CD4[+] T cells are sustained by site-specific levels of self-renewal and continuous replacement

Jodie Chandler[1†], M Elise Bullock[2†], Arpit C Swain[2], Cayman Williams[1], Christiaan H van Dorp[2], Benedict Seddon[1*], Andrew J Yates[2*]

[1]Institute of Immunity and Transplantation, Division of Infection and Immunity, UCL, Royal Free Hospital, London, United Kingdom; [2]Department of Pathology and Cell Biology, Columbia University Irving Medical Center, New York, United States

## eLife Assessment

This article provides a **compelling** and rigorous quantitative analysis of the turnover and maintenance of CD4[+] tissue-resident memory T cell clones, in the skin and the lamina propria. It provides a **fundamental** advance in our understanding of CD4 T cell regulation. Interestingly, in both tissues, maintenance involves an influx from progenitors on the time scale of months. The evidence that is based on fate mapping and mathematical inference is strong, although open questions on the interpretation of the Ki67-based fate mapping remain.

**Abstract** Tissue-resident memory T cells ($T_{RM}$) protect from repeat infections within organs and barrier sites. The breadth and duration of such protection are defined at minimum by three quantities: the rate at which new $T_{RM}$ are generated from precursors, their rate of self-renewal, and their rate of loss through death, egress, or differentiation. Quantifying these processes individually is challenging. Here we combine genetic fate mapping tools and mathematical models to untangle these basic homeostatic properties of CD4[+] $T_{RM}$ in the skin and gut lamina propria (LP) of healthy adult mice. We show that CD69[+]CD4[+] $T_{RM}$ in skin reside for ~24 days and self-renew more slowly, such that clones halve in size approximately every 5 weeks, and approximately 2% of cells are replaced daily from precursors. CD69[+]CD4[+] $T_{RM}$ in LP have shorter residencies (~14 days) and are maintained largely by immigration (4–6% per day). We also find evidence that the continuous replacement of CD69[+]CD4[+] $T_{RM}$ at both sites derives from circulating effector-memory CD4[+] T cells, in skin possibly via a local CD9[−] intermediate. Our approach maps the ontogeny of CD4[+] $T_{RM}$ in skin and LP and exposes their dynamic and distinct behaviours, with continuous seeding and erosion potentially impacting the duration of immunity at these sites.

## Introduction

Resident memory T cells ($T_{RM}$) provide immune surveillance and protection in tissues throughout the body (*Szabo et al., 2019*), but the mechanisms by which they are maintained are not well understood. Conventional CD4[+] and CD8[+] $T_{RM}$ in mice and humans are not intrinsically long-lived, but appear to self-renew slowly as assessed by readouts of cell division such Ki67 expression or BrdU incorporation, at levels that vary across tissues (*Gebhardt et al., 2009*; *Watanabe et al., 2015*; *Park et al., 2018*; *Strobl et al., 2020*; *Divito et al., 2020*; *Christo et al., 2021*). After infection or immune challenge, the numbers of elicited $T_{RM}$ may also be sustained by influx from precursor populations, although the extent to which this occurs is unclear and is likely also cell subset- and tissue-dependent. For example,

*For correspondence:
benedict.seddon@ucl.ac.uk (BS);
andrew.yates@columbia.edu
(AJY)

[†]These authors contributed equally to this work

Competing interest: The authors declare that no competing interests exist.

in the lung there is evidence both for (*Zammit et al., 2006*; *Ely et al., 2006*; *Slütter et al., 2017*; *Van Braeckel-Budimir et al., 2018*; *Takamura and Kohlmeier, 2019*) and against (*Takamura et al., 2016*; *Bullock et al., 2024*, *van Dorp et al., 2025*) ongoing recruitment of new $T_{RM}$ following respiratory virus infections. Within skin, $T_{RM}$ may be renewed or supplemented slowly from precursors in the setting of graft-versus-host disease (*Divito et al., 2020*), and from circulating central memory $T_{CM}$ or effector memory $T_{EM}$ following infection or sensitisation (*Gaide et al., 2015*; *Matos et al., 2022*). In the small intestine, however, resident $CD4^+$ and $CD8^+$ $T_{RM}$ appear to persist for months to years with slow self-renewal without appreciable influx (*Bartolomé-Casado et al., 2019*; *Bartolomé-Casado et al., 2021*).

The dynamics of production and loss of $T_{RM}$ in the steady state are even less well understood, and measuring these processes is important for several reasons. The balance of loss and self-renewal defines the persistence of clonal populations and hence the duration of protective immunity. Further, while self-renewal can at best preserve clonal diversity within a tissue site, any supplementation or replacement by immigrant $T_{RM}$ will perturb the local TCR repertoire. In particular, any significant influx into $T_{RM}$ niches in the absence of overt infection may be a competitive force, potentially reducing the persistence of $T_{RM}$ previously established in response to infection or challenge.

The kinetics of circulating memory T cells have been quantified extensively in both mice and humans using dye dilution assays (*Choo et al., 2010*), deuterium labelling (*Westera et al., 2013*; *Westera et al., 2015*; *Costa Del Amo et al., 2018*; *Baliu-Piqué et al., 2019*; *Baliu-Piqué et al., 2018*, *van den Berg et al., 2021*), and BrdU labelling, either alone (*Younes et al., 2011*; *Ganusov and De Boer, 2013*) or in combination with fate reporters (*Gossel et al., 2017*; *Hogan et al., 2019*; *Bullock et al., 2024*) or T cell receptor excision circles (*den Braber et al., 2012*). Using mathematical models to interpret these data, these studies identified rates of production, cellular lifespans, and signatures of heterogeneity in turnover. Modelling has also established evidence for continuous replenishment of circulating memory $CD4^+$ T cells from precursors throughout life in specific-pathogen-free mice, driven by a combination of environmental, commensal, and self antigens (*Gossel et al., 2017*; *Hogan et al., 2019*; *Bullock et al., 2024*). Quantification of $T_{RM}$ dynamics has to date been restricted largely to measuring the net persistence of $CD8^+$ $T_{RM}$ following infection in mice in a variety of tissues (*Morris et al., 2019*; *Wijeyesinghe et al., 2021*). Far less is known regarding $CD4^+$ $T_{RM}$, which typically outnumber their $CD8^+$ counterparts (*Szabo et al., 2019*), and there have been very few attempts to dissect the kinetics of either subset (*Bullock et al., 2024*, *van Dorp et al., 2025*).

In general, measuring these basic parameters in isolation is challenging, partly due to their sensitivity to assumptions made in the models (*De Boer and Perelson, 2013*), but also because division-linked labelling alone may not distinguish in situ cell division and the supplementation of a population from labelled precursors. These issues can be addressed by modelling different readouts of cell fate simultaneously (*Bains et al., 2009*; *den Braber et al., 2012*; *Costa Del Amo et al., 2018*; *De Boer and Yates, 2023*; *Bullock et al., 2024*).

With these challenges in mind, here we integrated data from two independent inducible fate reporter systems to study $CD4^+$ $T_{RM}$ homeostasis in mice. Each system allows one to track the fates of defined populations of cells and their descendants. One labels all $CD4^+$ T cell subsets at any given moment, which effectively provides an age 'timestamp'. The other labels cells that are dividing during a given time window. In combination, these systems allowed us to establish a quantitative model of the basal homeostatic properties of $CD4^+$ $T_{RM}$ within the skin and the lamina propria (LP) of the small intestine in healthy mice. In particular, we could unpick the contributions of self renewal and de novo cell production that underpin their maintenance, and explore their relationships to circulating T cell subsets.

## Results
### Combining cell fate reporters and models to measure $T_{RM}$ replacement, loss, and self-renewal

To study the homeostatic dynamics of tissue-resident $CD4^+$ memory T cells in healthy mice, we used in concert two genetic fate mapping tools in which cohorts of peripheral T cells and their offspring can be induced to express permanent fluorescent markers (*Figure 1A*). These reporter strains were previously used separately to study the turnover of naive and circulating memory T and B cells (*Verheijen et al.,*

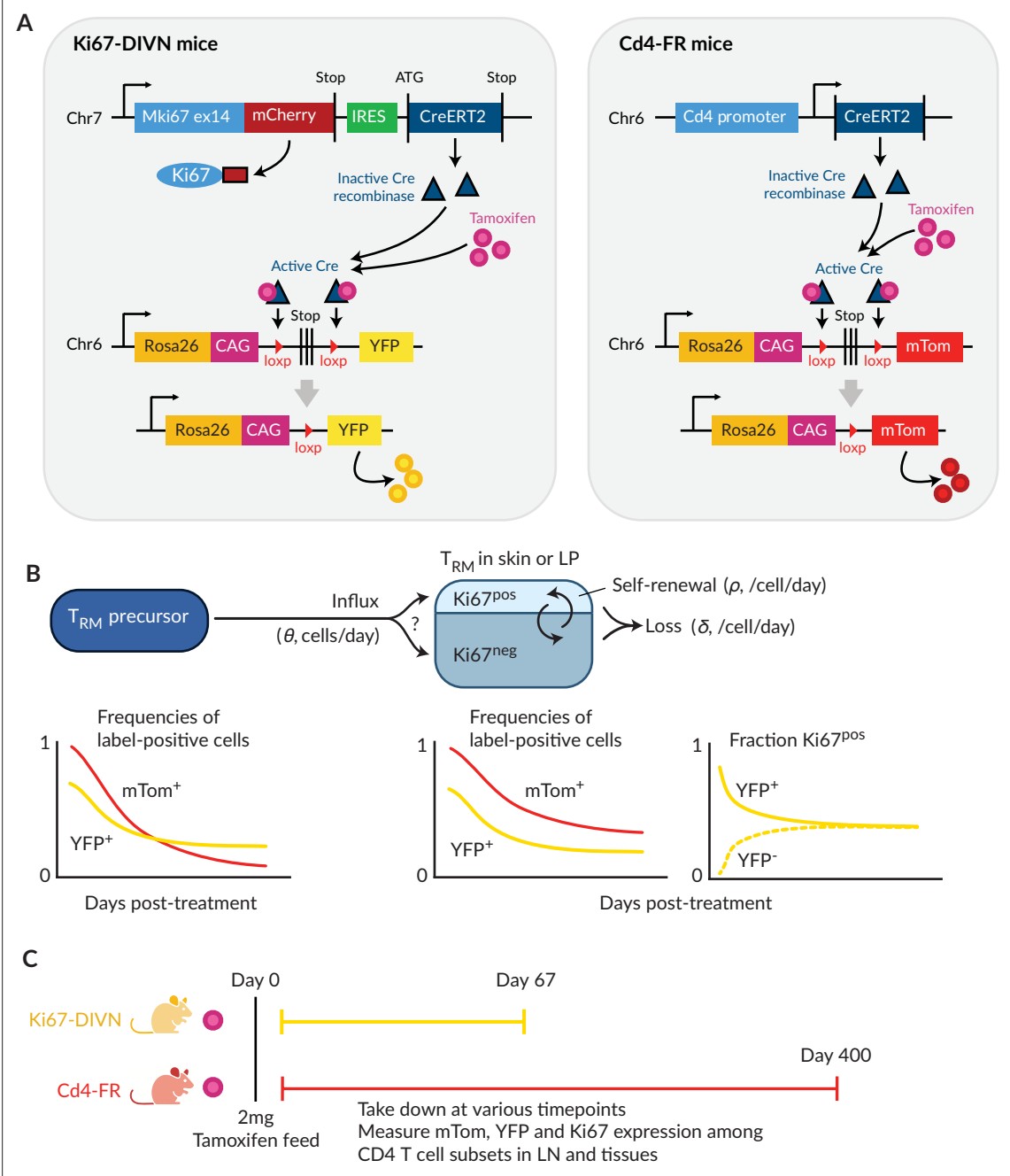

**Figure 1.** Overview of the fate-reporter approach to quantifying the homeostasis of skin- and lamina propria (LP)-resident CD4⁺ T cells. (**A**) Ki67 and CD4 reporter constructs. (**B**) Schematic of a simple mathematical model of $T_{RM}$ homeostasis at steady state. New cells enter a $T_{RM}$ subset in skin or LP (the 'target' population) from a precursor population at rate $\theta$ (cells per unit time). YFP and mTom expression among immigrant cells is assumed to be that of the precursor population in bulk. We considered three possibilities for the Ki67 expression levels among new immigrants: a 'quiescent' mode (all Ki67$^{low}$), 'neutral' recruitment (Ki67$^{high}$ frequency identical to that of the precursor), or 'division-linked' (all recruited cells Ki67$^{high}$). The target cells are assumed to be a kinetically homogeneous population that self-renews at average rate $\rho$, such that their mean interdivision time is $1/\rho$. Cells are also lost through death, differentiation or tissue egress at total rate $\delta$, implying a mean residence time of $1/\delta$. The combination of these processes determines the time courses of frequencies of YFP⁺ and mTom⁺ cells with the target population. (**C**) Experimental design.

The online version of this article includes the following figure supplement(s) for figure 1:

**Figure supplement 1.** Flow cytometric analysis of tissue-resident T cell subsets.

*2020*; *Lukas et al., 2023*; *Bullock et al., 2024*). In the Ki67[mCherry-CreERT] Rosa26[RcagYFP] system, henceforth Ki67-DIVN, fluorescent reporters are linked to the expression of Ki67, a nuclear protein expressed during cell division and for 3–4 days afterwards (*Gossel et al., 2017*; *Miller et al., 2018*). Specifically, these mice express both a Ki67-mCherry fusion protein and inducible CreERT from the *Mki67* locus, together with a *Rosa26[RcagYFP]* Cre reporter construct. Treatment of mice with tamoxifen therefore induces YFP in cells expressing high levels of Ki67, and YFP is then stably expressed by these cells and their offspring. Expression of Ki67-fused mCherry gives a constitutive live readout of Ki67 expression, independent of tamoxifen treatment and YFP expression. In the second fate reporter, CD4[CreERT] Rosa26[RmTom] mice (Cd4-FR), the Cre reporter is constructed such that cells expressing CD4 during tamoxifen treatment permanently and heritably express the fluorescent reporter mTomato (mTom).

In a closed population of cells at steady state, self-renewal must be balanced by loss and so, following tamoxifen treatment, the frequencies of cells expressing YFP or mTom within any such population would remain constant. Therefore, any decline in the frequency of either reporter among $T_{RM}$ after treatment must derive from the influx of label-negative cells from an upstream (precursor) population. In the Ki67-DIVN mice, these will be descendants of cells that were not dividing at the time of tamoxifen treatment; in the Cd4-FR mice, labelled CD4[+] cells will slowly be replaced by the descendants of those generated in the thymus after treatment. The shape of this decline will be determined by the combination of the net loss rate of $T_{RM}$ from the tissue (through death, egress, or differentiation, offset by any self-renewal) and the label content of immigrant $T_{RM}$ (*Figure 1B*). To refer to the persistence of individual $T_{RM}$ cells, we will use the term 'residence time' rather than 'lifespan' to reflect the multiple potential mechanisms of loss from tissues.

To quantify these processes, we treated cohorts of both reporter mice, aged between 4 and 15 weeks, with a single 2 mg pulse of tamoxifen (*Figure 1C*). Over 9-week (Ki67 reporter) and 57-week (CD4 reporter) chase periods, we measured the frequencies of labelled cells among antigen-experienced CD4[+] T cell subsets isolated from skin and the LP of the small intestine, and within circulating naive and memory T cell subsets derived from lymph nodes (*Figure 1—figure supplement 1A*). By combining these frequencies with measures of Ki67 expression and describing the resulting set of time series with simple mathematical models, we aimed to estimate the basic parameters underlying $T_{RM}$ kinetics.

## CD4[+] $T_{RM}$ in skin and lamina propria are continuously replaced from precursors

We considered two populations within both skin and LP, identified as tissue-localised by virtue of their protection from short-term in vivo labelling ('Methods' and *Figure 1—figure supplement 1B*). One was effector-memory (EM) phenotype (CD4[+]CD44[hi] CD62L[lo]) T cells in bulk, which we studied in order to gain the broadest possible picture of memory T cell dynamics at these sites. We also considered the subset of these cells that expressed CD69, a canonical marker of CD4[+] T cell residency across multiple tissues (*Szabo et al., 2019*). We saw no significant changes with mouse age in the numbers of either population within skin (*Figure 2A*, $p > 0.67$) or LP (*Figure 2B*, $p > 0.39$). There were also no significant changes in any of these quantities with time since tamoxifen treatment ($p > 0.24$). We therefore assumed that the skin- and LP-localised T cell subsets we considered were at, or close to, homeostatic equilibrium during the chase period. For brevity, we refer to tissue-localised CD4[+]CD44[hi]CD62L[lo] T cells in bulk as EM, and their CD69[+] subset as $T_{RM}$.

During the first few days after tamoxifen treatment, YFP and mTom expression increased continuously within the skin and LP subsets (*Figure 2C*), as well as among CD4[+] naive (CD44[lo] CD62L[hi]), central memory ($T_{CM}$, CD44[hi] CD62L[hi]), and effector memory ($T_{EM}$, CD44[hi] CD62L[lo]) T cells recovered from lymph nodes (*Figure 2D*). These initial increases were driven in part by the intracellular dynamics of the induction of the fluorescent reporters. We therefore began our analyses at day 5 post-treatment, by which time induction was considered complete and the subsequent trajectories of label frequencies reflected only the dynamic processes of cell production and loss. A key observation was that mTom expression within the skin and LP subsets then declined slowly (roughly seven- to eight-fold over the course of a year, *Figure 2C*), indicating immediately that these populations were being continuously replaced from precursors. Early in the chase period YFP[+] $T_{RM}$ expressed Ki67 at higher levels than YFP[−] cells, as expected, but Ki67 expression in the two populations converged at later times. We return to the interpretation of these kinetics below.

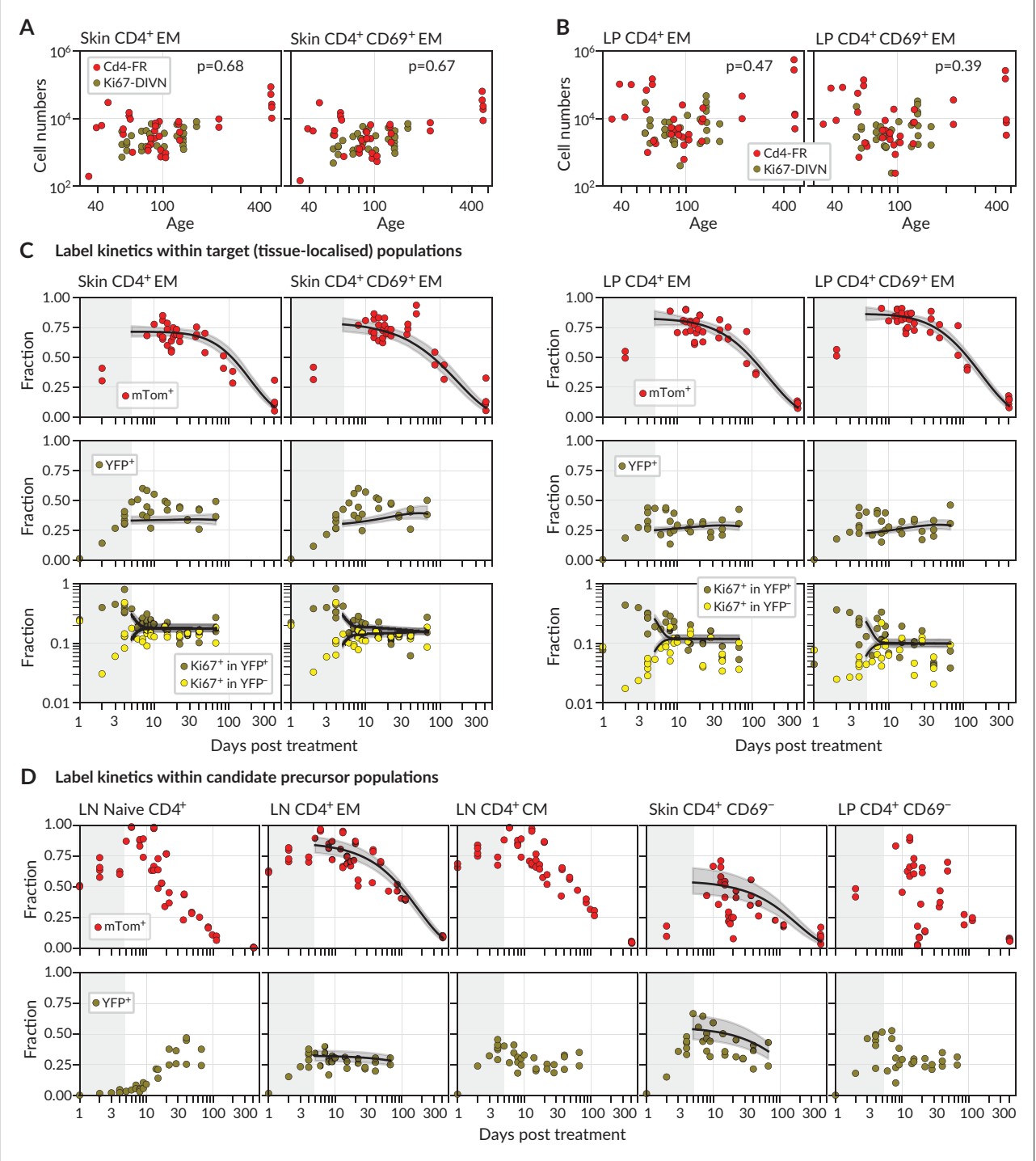

**Figure 2.** Modelling fate reporter dynamics among CD4⁺ T cell subsets in skin and lamina propria (LP). (**A, B**) Numbers of CD4⁺ effector-memory (EM) phenotype and CD4⁺CD69⁺ T cells recovered from (**A**) ear skin and (**B**) LP in the small intestine, against mouse age. p-values are derived from Spearman rank correlation on data from the Cd4-FR and Ki67-DIVN strains combined. (**C**) Observed (points), best-fit model trajectories (black lines), and 95% credible intervals (grey envelopes) of the frequencies of mTom⁺ and YFP⁺ CD4⁺ T cell subsets within skin and LP, and the proportions of YFP⁺ and YFP⁻ cells that expressed Ki67, with time since tamoxifen treatment. Shaded rectangles indicate the 5-day period over which reporter expression was being induced; data from these periods were not used in fitting. (**D**) Frequencies of mTom⁺ and YFP⁺ cells within lymph node CD4⁺ T cell subsets, and within CD4⁺CD44⁺CD69⁻ T cells in skin and LP. Overlaid are the empirical descriptions of these trajectories from the best-fitting model in which that population was identified as a precursor (Appendix 1 section 'Model fitting'); grey envelopes are 95% credible intervals. (The naive and CM in lymph nodes, and CD69⁻ cells in LP, were never identified as favoured precursors.) Number of Cd4-FR mice: 36; number of Ki67-DIVN mice: 31.

*Figure 2 continued on next page*

*Figure 2 continued*

The online version of this article includes the following figure supplement(s) for figure 2:

**Figure supplement 1.** Prior and posterior distributions of parameters for the best-fitting models.

**Figure supplement 2.** Fits of top-, second-, and lowest-ranked models for each target population.

We then investigated the extent to which the simple model illustrated in *Figure 1B* could explain these trajectories. Given the observation of continued recruitment, any time variation in the label content of T$_{RM}$ precursors might leave an imprint on the label kinetics of skin or LP T$_{RM}$ themselves and thereby help us identify their developmental pathways. We reasoned that plausible T$_{RM}$ precursors were LN-derived CD4$^+$ naive, T$_{CM}$ or T$_{EM}$; we also considered the possibilities that CD69$^-$ cells within skin and LP are the direct precursors of the local CD69$^+$ populations. Therefore, we used empirical functions to describe the time courses of the frequencies of YFP$^+$ and mTom$^+$ cells within these populations (*Figure 2D*) and used these to represent the label composition of cells entering the tissue subsets.

For each tissue subset ('target') and precursor pair, we fitted the model simultaneously to six time courses; the frequencies of (i) YFP expression and (ii) mTom expression among target cells, the proportions of Ki67$^{high}$ cells among (iii) YFP$^+$ and (iv) YFP$^-$ target cells, and the (v) YFP and (vi) mTom expression kinetics within the precursor ('Methods' and Appendix 1). For each precursor/target pair, we considered three modes of influx – one in which new immigrant T$_{RM}$ are Ki67$^{low}$ ('quiescent' recruitment); another in which their Ki67 expression directly reflects that of the precursor ('neutral' recruitment); and a third in which immigrants have recently divided (Ki67$^{high}$), perhaps through an antigen-driven process ('division-linked' recruitment).

## Skin and LP CD4$^+$ T$_{RM}$ have similar residence times but exhibit distinct contributions of replacement and self-renewal

For each combination of target population, potential precursor, and potential mode of recruitment, we were able to estimate rates of influx, mean residence times, and mean interdivision times for the target population (*Figure 3* and *Appendix 1—table 1*; prior and posterior distributions of the parameters of the best-fitting models are shown in *Figure 2—figure supplement 1*). The mean residence times of both EM and T$_{RM}$ within skin and LP were ~3 weeks and 2 weeks, respectively. The means of production of new cells differed at the two sites, however. In skin, around 2% of both populations were replaced daily by influx, comparable to the rates of constitutive replacement of circulating memory CD4$^+$ T cell subsets (*Gossel et al., 2017*; *Hogan et al., 2019*; *Bullock et al., 2024*), and EM and T$_{RM}$ self-renewed every 6 and 7 weeks, respectively. In contrast, within LP these subsets divided less often (every 7–9 weeks) and relied on higher levels of recruitment (4–6% per day) for their maintenance.

From these basic quantities, we could derive several other important measures of T$_{RM}$ behaviour. First, the balance of the rates of loss ($\delta$) and self-renewal ($\rho$) defines the persistence of a cohort of T cells, which is distinct from the lifespan of its constituent cells (*Costa Del Amo et al., 2018*; *De Boer and Yates, 2023*). Specifically, the quantity $\ln(2)/(\delta - \rho)$ is the average time taken for a cohort and their descendents to halve in number. While we studied polyclonal populations here, this quantity applies equally well to measuring the persistence of a TCR clonotype, so we refer to it as a clonal half life (*Bullock et al., 2024*). The substantial rates of self-renewal in skin led to clonal half lives of just over a month. The lower levels of self-renewal and higher levels of replacement in LP resulted in shorter clonal half lives of approximately 2 weeks.

Importantly, our estimates of these quantities depended to varying degrees on the choice of precursor and mode of recruitment (*Figure 3*). For example, intuitively, given the observed level of Ki67 within each target population, the greater the levels of Ki67 within newly recruited cells, the less must derive from self-renewal within the tissue; hence, if one assumes that recruitment is division-linked, estimated division rates are reduced. Similarly, as discussed above, the label content of the precursor influences the net loss rate of label in the target, which was most clearly reflected in the loss of mTom$^+$ cells over the longer chase period (*Figure 2C and D*). For example, mTom was lost most rapidly within naive CD4$^+$ T cells (*Figure 2D*) due to the influx of label-negative cells from the thymus. Models in which naive T cells were the direct precursor of T$_{RM}$ therefore predicted greater clonal persistence within tissues.

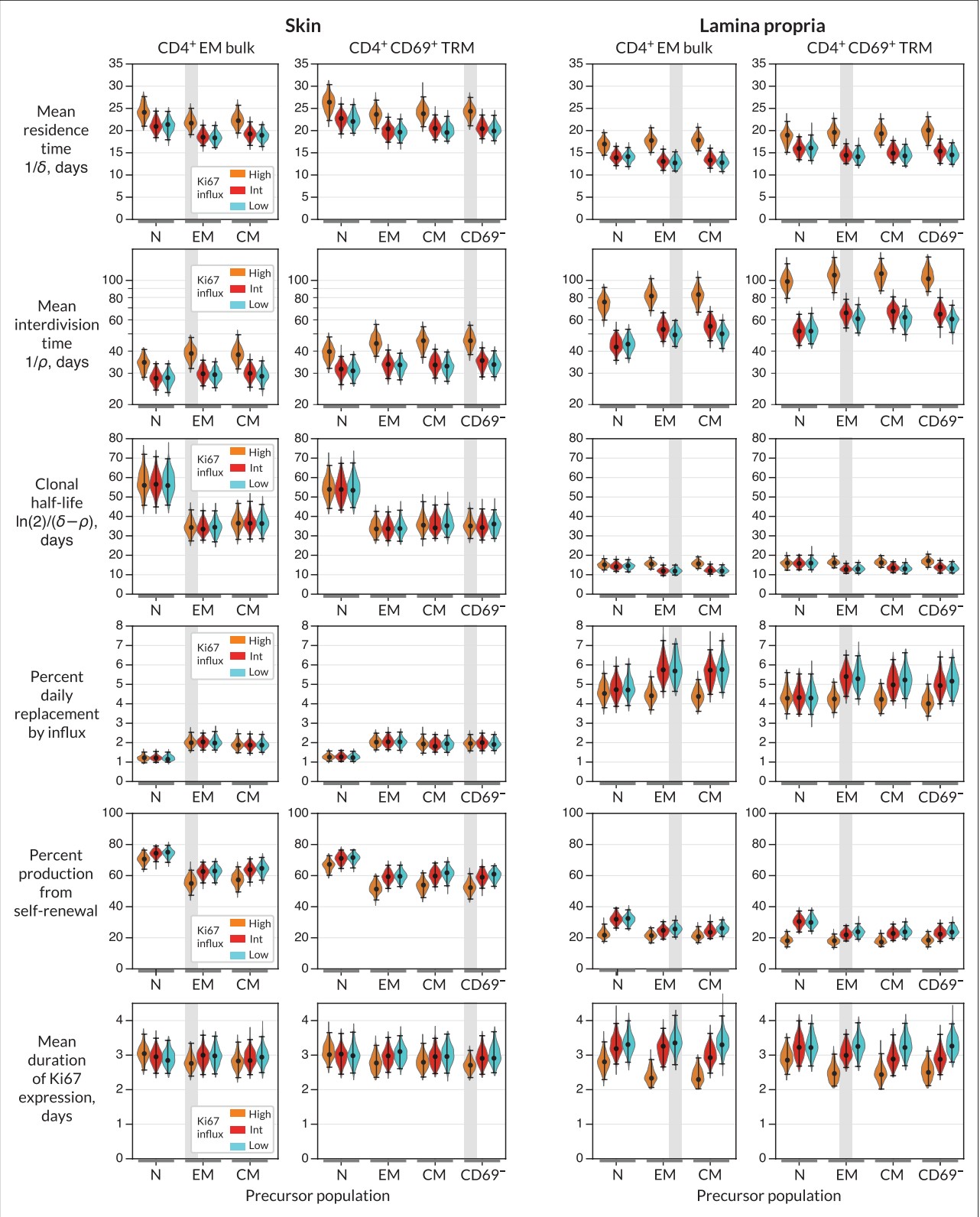

**Figure 3.** Parameters governing the homeostasis of antigen-experienced CD4+ T cells localised within skin and lamina propria (LP) in adult mice. Violin plots indicate the posterior distributions of parameters. Black points and bars; best (maximum a posteriori) estimates and 95% credible intervals. For each population (target) in skin or LP, potential precursors were from lymph nodes, CD4+ naive, central memory and effector memory T cells (N, EM, and CM); and for CD4+ $T_{RM}$, CD69+ $T_{RM}$ within the same tissue. For each precursor/target pair, we considered three potential levels of Ki67 expression on

*Figure 3 continued on next page*

immigrant cells (L-R; orange, red, and blue violin plots). 'High', the division-linked model (new cells enter as Ki67high); 'Int', neutral model (Ki67 expression among new cells mirrors that of the precursor); and 'Low', the quiescent model (new cells enter as Ki67low). Shaded regions highlight the parameter estimates derived from the favoured model (precursor and Ki67 levels on immigration) for each target (*Figure 4A*).

We saw very little decline in YFP expression levels during the 2-month chase period (*Figure 2C*) due in part to the sustained levels of YFP within the putative precursor populations (*Figure 2D*), which 'topped up' YFP-expressing $T_{RM}$. As a result, YFP kinetics within the tissues were not strongly informative regarding rates of replacement. However, the rate of convergence of Ki67 within YFP$^+$ and YFP$^-$ cells (*Figure 2C*) put clear constraints on the duration of Ki67 expression, which at approximately 3 days (*Figure 3*) was consistent with previous estimates. This quantity in turn was informative for estimating rates of self-renewal. Further, the observed convergence of Ki67 expresssion within these two subsets is consistent with the assumption of homogeneity in the rates of division and loss within skin and LP.

## CD4$^+$ CD69$^+$ T$_{RM}$ within LP likely derive predominantly from circulating T$_{EM}$ in lymph nodes, while those in skin may derive from a CD69$^-$ intermediate

To more precisely quantify the kinetics of each target population, we assessed the relative support for each combination of precursor and mode of recruitment (*Figure 4A*). Each of these weights summarises the magnitude and uncertainty of a model's out-of-sample prediction error of the label kinetics within the target population ('Methods').

We found that the data were quite strongly informative regarding the immediate ancestors of tissue subsets. From the candidate set of models, the weighting strongly favoured lymph-node-derived EM as the closest precursor to CD4$^+$ EM within both skin and LP. However, within skin, local CD69$^-$ cells were the favoured precursor to CD69$^+$ T$_{RM}$. The evidence was generally more equivocal regarding the

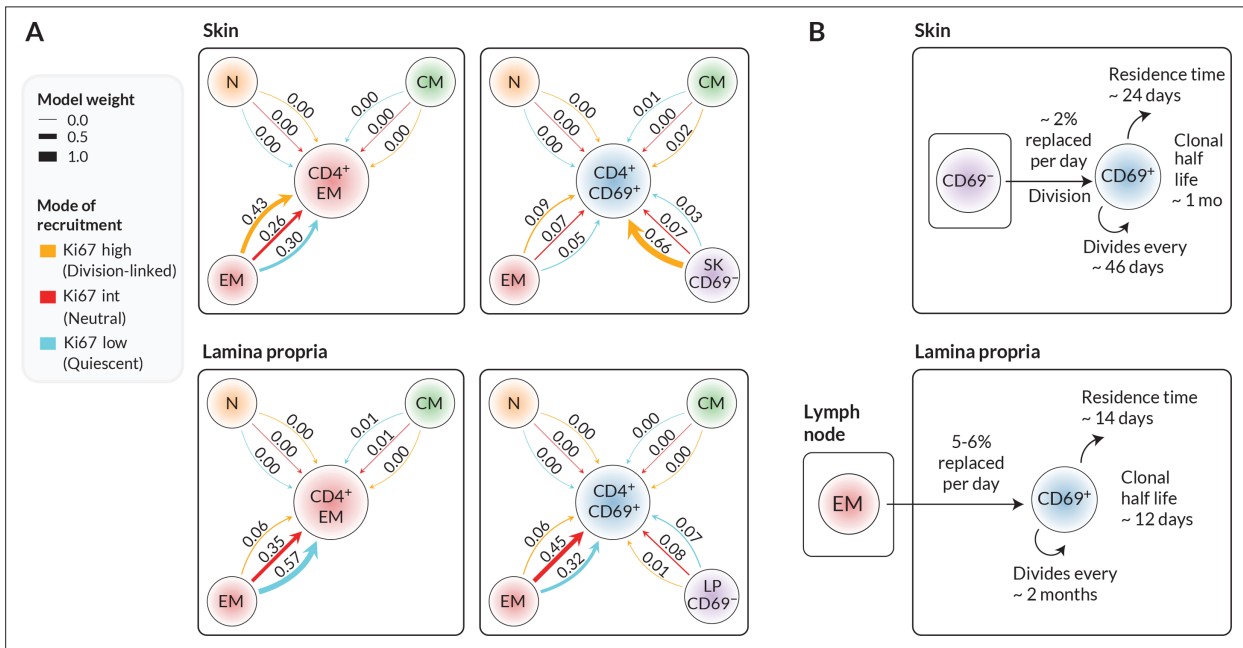

**Figure 4.** Candidate ontogenic pathways of CD4$^+$ CD69$^+$ T$_{RM}$ in skin and lamina propria (LP). (**A**) For each target population with skin and LP (CD4$^+$ EM bulk and CD4$^+$CD69$^+$ T cells), we calculated the relative support for alternative pathways and modes of recruitment using LOO-IC weights ('Methods'). For potential precursors, N, CM, and EM refer to naïve, T$_{CM}$, and T$_{EM}$ in lymph nodes; for CD69$^+$ targets we also considered CD69$^-$ cells within the same tissue as a possible precursor. As described in the text, for each precursor/target pair, there were three potential submodels relating to the possible levels of Ki67 on newly recruited cells. Model weights sum to 1 for each target population, with the width of each arrow reflecting a model's degree of support. (**B**) Schematics of the most strongly supported pathways of development of CD4$^+$CD69$^+$ T$_{RM}$ in skin and LP, with approximate values of key kinetic parameters (*Figure 3* and *Appendix 1—table 1*).

mode in which cells are recruited into skin and LP, although for skin we saw substantial evidence (66% of model support) for a division-linked transition from CD69$^-$ to CD69$^+$ cells. *Figure 4B* summarises the developmental trajectories and kinetics of T$_{RM}$ in skin and LP that were supported most strongly by our analyses. Parameter estimates and credible intervals for these models are highlighted with vertical shaded regions in *Figure 3* and are detailed in *Appendix 1—table 1*. Visual differences between models are shown in *Figure 2—figure supplement 2*, where for each target population we overlay the fits from top-ranked, second-ranked, and lowest-ranked models.

## Validation of residence times using Ki67 expression directly

As a consistency check, when a population is at or close to steady state, bounds on the mean residence time of cells can be estimated using only the measured frequency of Ki67 within the target population, and the daily rate of replacement (Appendix 1 section 'Estimating average cell residence times using Ki67'). In skin, both EM and T$_{RM}$ are replaced at the rate of 2% per day and have Ki67 expression frequencies of around 0.15 (*Figure 2C*). The approximation then yields residence times in the range 22–28 days, depending on whether immigrant T$_{RM}$ are Ki67$^{low}$ or Ki67$^{high}$, respectively. In LP, with 5.5% daily replacement and Ki67 frequencies of around 0.07, we estimate residence times of 14–25 days. Both estimates are in good agreement with those from the model fitting. Further validation of these results, and dissection of the kinetics, might be achieved by manipulating cell trafficking, although this would potentially impact multiple processes at once. For example, treating mice with the sphingosine 1-phosphate receptor agonist FTY720 would block tissue ingress and egress. This would leave self-renewal as the only means of T$_{RM}$ production and would also remove the component of the loss rate that is due to cells leaving the tissue. In principle, one could then gain estimates of the intrinsic lifespan of T$_{RM}$, rather than their tissue residence time. However, parameter estimation would then require accurate measurements of cell numbers within the tissue.

## Discussion

Our analysis indicates that CD4$^+$ T$_{RM}$ are not intrinsically long-lived, but instead are sustained by both self-renewal and supplementation from circulating precursors. By combining fate reporting methods with mathematical models, we also showed that it is possible to separately quantify the processes that underlie their persistence. We saw quite distinct contributions of recruitment and self-renewal of both subsets within skin and LP. The basis of this difference is unclear, but we speculate that the large antigenic burden within the small intestine drives the higher levels of T$_{RM}$ recruitment and clonal erosion within the LP. We showed that estimates of these quantities depend on the identity of the immediate precursor, whose label kinetics propagate downstream into the population of interest, and the extent of any cell division that occurs around the time of differentiation or ingress. However, by using easily interpretable mathematical models we were able to measure the support for different pathways and modes of recruitment into each subset.

The schematic in *Figure 1B* illustrates a hypothetical example in which the frequency of YFP-expressing cells within a precursor declines. This trend is then reflected downstream in the target. However, in our experiments the kinetics of YFP in most of the putative precursors were quite flat (*Figure 2D*). As noted above, these kinetics were likely due largely to the continued influx of new cells into circulating memory subsets, in turn likely from naive precursors (*Gossel et al., 2017*; *Hogan et al., 2019*; *Bullock et al., 2024*). These had increasing levels of YFP (*Figure 2D*, left panel) deriving from thymocytes that were dividing rapidly during treatment. YFP levels among naive T cells were also probably sustained to an extent by low-level residual labelling of thymic progenitors (*Lukas et al., 2023*). YFP expression was therefore not cleanly 'washed out' in the periphery. The data from the labelling of CD4-expressing cells were more informative for dissecting turnover; mTom$^+$ cells were clearly diluted out of all peripheral populations by the descendants of mTom$^-$ thymocytes.

In these reporter mice, YFP and mTom were induced quickly in all subsets to different degrees; therefore, our inferences regarding precursor–target relationships were not informed by the initial levels of label in each. (For example, imagine a rapidly dividing target fed by a slowly dividing precursor; initially, YFP levels in the target would be higher than those in the precursor.) The hierarchy of levels of label in different subsets *would* be informative if one expects targets to begin with no label at all; for instance, in the busulfan chimeric mouse system (*Hogan et al., 2015*) new, thymically

derived 'labelled' (donor) cells progressively infiltrate replete 'unlabelled' (host) populations. In that case, one can immediately reject certain differentiation pathways by examining the sequence of accrual of donor cells in different subsets. In the systems we use here, information regarding lineage relationships is contained instead in the trends in YFP and mTom frequencies after treatment because precursor kinetics must leave an imprint on the target (*Figure 1B*). This information is particularly useful if two populations exhibit opposing trajectories – they are then unlikely to be immediately related.

On a technical note, in general one can reduce experimental variation by comparing quantities derived from the same individual. We showed previously that in some situations exploiting the within-mouse grouping of observations can reduce uncertainty and refine parameter estimates when modelling cell dynamics (*Yates et al., 2007*). We were not able to take this approach here, however, because the frequency of cells expressing YFP or mTom within a given subset in a particular mouse depends on the accumulated history of label in that subset's precursor. We were unable to identify these trajectories for each animal, so we were obliged to use population averages (*Figure 2D*). To mitigate any biases this averaging might introduce, we fitted these empirical functions simultaneously with the models of label kinetics in the targets. This conservative strategy propagated the uncertainty in the precursor trajectories into our conclusions.

Our analysis was rooted in the observation that all CD4$^+$ T cell subsets within these tissues were at steady state. This dynamic equilibrium dictates that immigration of new cells must be accompanied by loss of existing ones. In the SPF mice studied here, these dynamic populations are likely specific for self or commensal antigens that are continuously expressed. It is possible that residence and interdivision times are distinct for $T_{RM}$ that might not be replenished long-term from precursors, such as those generated in acute infections. Further, our simple model can explain the stable maintenance of $T_{RM}$ numbers in healthy skin and LP without needing to invoke the concept of a homeostatic niche, such as a competitive limit to cell densities. In this model, any increase or decrease in the rate of influx into a tissue will simply lead to a new equilibrium at higher or lower cell densities, respectively. Indeed repeated vaccinia virus challenges can drive progressive increases in the number of virus-specific CD8$^+$ $T_{RM}$ in skin that are detectable for months (*Jiang et al., 2012*), and heterologous challenges appear not to erode pre-existing LCMV-specific CD8$^+$ $T_{RM}$ (*Wijeyesinghe et al., 2021*). However, whether the same flexibility manifests among CD4$^+$ $T_{RM}$ following repeated challenges is unclear.

Our goal here was to assess the support for external replenishment of effector-memory-like CD4$^+$ T cells in bulk and the dominant CD69$^+$ $T_{RM}$ subset. We found evidence that CD69$^+$CD4$^+$ $T_{RM}$ in skin derive at least in part from a local CD69$^-$ precursor. With richer phenotyping of tissue-localised cells, we could in principle use labelling trajectories to define more fine-grained differentiation pathways. One issue is that accurate measurement of label frequencies becomes more difficult as one resolves $T_{RM}$ into smaller subsets; indeed, we saw that label kinetics within the small CD69$^-$ populations were relatively noisy. Another issue is that label kinetics operate on the timescales of the net loss rate of cell populations – death and onward differentiation, balanced by self-renewal. More frequent sampling would be required to resolve more transitory intermediates. Nevertheless, our study clearly exposes the highly dynamic nature of CD4$^+$ $T_{RM}$, sustained throughout life by both self-renewal and continued influx from precursors. This tissue-specific influx, particularly if there are competitive limits to $T_{RM}$ occupancy, may contribute to the differential longevity of immunity at different barrier sites.

## Methods
### Reporter mouse strains
Ki67$^{mCherry-CreERT}$ Rosa26$^{RcagYFP}$ (Ki67-DIVN) and CD4$^{CreERT}$ Rosa26$^{RmTom}$ (Cd4-FR) mice have been described previously (*Bullock et al., 2024*). Experimental Ki67-DIVN mice were homozygous for indicated mutations at both the *Ki67* and *Rosa26* loci. Experimental Cd4-FR mice were heterozygous for the indicated mutations at both the *Cd4* and *Rosa26* loci. Tamoxifen (Sigma) was diluted to 20 mg/mL in corn oil (Fisher Scientific) and 100 µl (2 mg) was administered to mice via oral feeding on day 0. Ki67-DIVN mice were injected with 2 µg Thy1.2-BV510 (53-2.1) (BioLegend) 3 min prior to sacrifice to label T cells in the circulation. This protocol typically achieves >99% staining of circulating cells (*Anderson et al., 2014*), and less than 3% of cells recovered from our tissue samples were label positive (*Figure 1—figure supplement 1B*), supporting the assumption of very low rates of false-positive

and false-negative events. Mice were subsequently taken down at specified timepoints post tamoxifen treatment for organ collection. All mice were bred in the Comparative Biology Unit of the Royal Free UCL campus and at Charles River laboratories, Manston, UK. Animal experiments were performed according to institutional guidelines and Home Office regulations under project licence PP2330953.

## Cell preparation

All peripheral lymph nodes (LNs), the small intestine (SI), and ear skin were taken from mice and processed into single-cell suspensions. LNs were mashed through two pieces of fine gauze in a Petri dish and washed with complete RPMI (Thermo Fisher) supplemented with 5% FCS (Thermo Fisher) (cRPMI). Cells were resuspended in cold PBS and counted using the CASY counter (Cambridge Bioscience). Peyer's patches were excised from the antimesenteric side of the SI before being opened longitudinally and SI contents scraped out. SI pieces were placed in 20 mL pre-warmed extraction media (cRPMI + 10 mM HEPES [Thermo Fisher] +5 mM EDTA [Sigma]+1 mM DTT [Abcam]) and incubated in 37 °C shaking incubator for 30 min at 200 rpm. Cells were filtered over 70 μm cell strainer (supernatant containing intra-epithelial lymphocytes not used), and SI pieces were placed in cold cRPMI supplemented with 10 mM HEPES and allowed to settle. The supernatant was carefully poured off and SI pieces were finely minced, added to 20 mL pre-warmed digestion media (RPMI + 10% FCS + 1.5 mg/mL collagenase VIII [Sigma]) and incubated in 37 °C shaking incubator for 30 min at 200 rpm. After digestion, cells were passed through 70 μm cell strainer and washed with cRPMI + 10 mM HEPES. The resulting cell suspension contains cells from the LP of the SI. Ear skin was excised and separated into dorsal and ventral sides. Skin was finely minced, added to 4 mL digestion buffer cRPMI + 50 mM HEPES + 37.5 μg/mL Liberase TL [Merck] + 3.125 mg/mL collagenase IV [Thermo Fisher] +1 mg/mL DNAse I [Merck] and incubated in 37 °C shaking incubator for 2 h at 200 rpm. Cells were filtered over 70 μm cell strainer and washed through with cRPMI.

## Flow cytometry

All cells isolated from skin and LP, and $5 \times 10^6$ LN cells were stained for analysis by flow cytometry. Cells were stained in 100 μl PBS with combinations of CD8α-BUV395 (53-6.7), CD25-BUV395 (PC61), CD62L-BUV737 (MEL-14), TCRγδ-BV421 (GL3), CD103-BV786 (M290) (all BD Biosciences); CD25-BV650 (PC61), CD103-BV421 (2E7), CD8α-BV570 (53-6.7), TCRγδ-BV605 (GL3), NK1.1-BV650 (PK136), CD44-BV785 (IM7), CD45.1-BV605 (A20), CD4-BV711 (RM4-5), CD8b.2-APC (53-5.8) (all BioLegend); TCRβ-PerCPCy5.5 (104) (Cambridge Bioscience); CD44-APCef780 (IM7) (eBioscience); and CD3-APCef780 (2C11), CD3-biotin (145-2C11), CD69-PeCy7 (H1.2F3), CD45.2-AF700 (104), nearIR live/dead, blue live/dead, yellow live/dead (all Thermo Fisher). Cells were fixed for 20 min with IC fix (Invitrogen) and washed twice in FACS buffer (PBS + 0.1% BSA). Flow cytometric analysis was performed on either a Cytek Aurora spectral flow cytometer or a conventional BD LSR-Fortessa and analysed using FlowJo software (Treestar).

## Cell count calculations

Cell numbers in LN were calculated by dividing the event count in a target population by the event count of live cells, multiplied by the total live cells in LN prep determined by CASY counter. Sizes of LP and skin populations were calculated using AccuCount (Spherotech) counting beads that were spiked into the sample prior to acquisition, as per the manufacturer's instructions.

## Mathematical modelling and statistical analysis

We fitted simultaneously the mathematical model illustrated in *Figure 1B* and described in Appendix 1 sections 'The kinetics of mTom expression derived from Cd4-FR mice' and 'Modelling the label trajectories derived from Ki67-DIVN mice', to the timecourses of frequencies of YFP⁺, Ki67^high in YFP⁺, Ki67^high in YFP⁻, and mTom⁺ cells in the target populations; and empirical descriptor functions describing the trajectories of the frequencies of YFP⁺ and mTom⁺ cells within precursor populations (all data shown in *Figure 2C*). We used a Bayesian estimation approach using Python and Stan (*Stan Development Team, 2024*) to perform these model fits. Code and data used to perform model fitting, details of the prior distributions for parameters, and figure generation notebooks are available at https://github.com/elisebullock/CD4TRM; copy archived at *Bullock, 2025*. Models were ranked

based on the information criteria estimated using the Leave-One-Out (LOO) cross validation method (*Vehtari et al., 2017*). See Appendix 1 for full details.

## Acknowledgements

This work was supported by the National Institutes of Health (R01 AI093870 and U01 AI150680) and the Medical Research Council (MR/P011225/1).

## Additional information

### Funding

| Funder | Grant reference number | Author |
|---|---|---|
| National Institutes of Health | R01 AI093870 | Jodie Chandler<br>M Elise Bullock<br>Arpit C Swain<br>Cayman Williams<br>Christiaan H van Dorp<br>Benedict Seddon<br>Andrew J Yates |
| National Institutes of Health | U01 AI150680 | Jodie Chandler<br>M Elise Bullock<br>Arpit C Swain<br>Cayman Williams<br>Christiaan H van Dorp<br>Benedict Seddon<br>Andrew J Yates |
| Medical Research Council | MR/P011225/1 | Jodie Chandler<br>M Elise Bullock<br>Arpit C Swain<br>Cayman Williams<br>Christiaan H van Dorp<br>Benedict Seddon<br>Andrew J Yates |

The funders had no role in study design, data collection and interpretation, or the decision to submit the work for publication.

### Author contributions

Jodie Chandler, Data curation, Formal analysis, Investigation, Methodology; M Elise Bullock, Data curation, Software, Formal analysis, Investigation, Methodology; Arpit C Swain, Software, Formal analysis, Investigation; Cayman Williams, Investigation, Methodology; Christiaan H van Dorp, Software, Investigation, Methodology; Benedict Seddon, Conceptualization, Supervision, Funding acquisition, Investigation, Methodology, Project administration, Writing – review and editing; Andrew J Yates, Conceptualization, Formal analysis, Supervision, Funding acquisition, Investigation, Methodology, Writing – original draft, Project administration, Writing – review and editing

### Author ORCIDs

M Elise Bullock ⓘ https://orcid.org/0000-0002-7876-4938
Cayman Williams ⓘ https://orcid.org/0000-0002-0518-1379
Christiaan H van Dorp ⓘ https://orcid.org/0000-0002-7504-9947
Benedict Seddon ⓘ https://orcid.org/0000-0003-4352-3373
Andrew J Yates ⓘ https://orcid.org/0000-0003-4606-4483

### Ethics

All mice were bred in the Comparative Biology Unit of the Royal Free UCL campus and at Charles River laboratories, Manston, UK. Animal experiments were performed according to institutional guidelines and Home Office regulations under project licence PP2330953.

Reviewer #1 (Public review): https://doi.org/10.7554/eLife.104278.3.sa1

Reviewer #2 (Public review): https://doi.org/10.7554/eLife.104278.3.sa2

Author response https://doi.org/10.7554/eLife.104278.3.sa3

## Additional files

### Supplementary files

MDAR checklist

### Data availability

Code and data used to perform model fitting, details of the prior distributions for parameters, and figure generation notebooks are available at https://github.com/elisebullock/CD4TRM, copy archived at *Bullock, 2025*.

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

## Appendix 1

**Appendix 1—table 1.** MAP estimates and 95% credible intervals of parameters from the statistically favoured models.

LP: lamina propria; LN: lymph node.

| Parameter | Tissue | Target | Recruitment mode | Precursor | Estimate |
|---|---|---|---|---|---|
| Mean residence time (days) | | CD4$^+$ EM | Quiescent (Ki67$^{low}$) | LN CD4$^+$ EM | 13 (11, 15) |
| | LP | CD4$^+$CD69$^+$ T$_{RM}$ | Neutral (Ki67$^{int}$) | LN CD4$^+$ EM | 14 (13, 16) |
| | | CD4$^+$ EM | Division-linked (Ki67$^{hi}$) | LN CD4$^+$ EM | 22 (20, 24) |
| | Skin | CD4$^+$CD69$^+$ T$_{RM}$ | Division-linked (Ki67$^{hi}$) | Skin CD69$^-$ | 24 (22, 27) |
| Mean interdivision time (days) | | CD4$^+$ EM | Quiescent (Ki67$^{low}$) | LN CD4$^+$ EM | 49 (44, 57) |
| | LP | CD4$^+$CD69$^+$ T$_{RM}$ | Neutral (Ki67$^{int}$) | LN CD4$^+$ EM | 66 (57, 75) |
| | | CD4$^+$ EM | Division-linked (Ki67$^{hi}$) | LN CD4$^+$ EM | 22 (20, 24) |
| | Skin | CD4$^+$CD69$^+$ T$_{RM}$ | Division-linked (Ki67$^{hi}$) | Skin CD69$^-$ | 24 (22, 27) |
| Clonal half-life (days) | | CD4$^+$ EM | Quiescent (Ki67$^{low}$) | LN CD4$^+$ EM | 10 (9, 12) |
| | LP | CD4$^+$CD69$^+$ T$_{RM}$ | Neutral (Ki67$^{int}$) | LN CD4$^+$ EM | 12 (10, 14) |
| | | CD4$^+$ EM | Division-linked (Ki67$^{hi}$) | LN CD4$^+$ EM | 34 (29, 41) |
| | Skin | CD4$^+$CD69$^+$ T$_{RM}$ | Division-linked (Ki67$^{hi}$) | Skin CD69$^-$ | 35 (30, 42) |
| Percent daily replacement by influx | | CD4$^+$ EM | Quiescent (Ki67$^{low}$) | LN CD4$^+$ EM | 5.7 (4.9, 6.9) |
| | LP | CD4$^+$CD69$^+$ T$_{RM}$ | Neutral (Ki67$^{int}$) | LN CD4$^+$ EM | 5.4 (4.5, 6.2) |
| | | CD4$^+$ EM | Division-linked (Ki67$^{hi}$) | LN CD4$^+$ EM | 2.0 (1.7, 2.4) |
| | Skin | CD4$^+$CD69$^+$ T$_{RM}$ | Division-linked (Ki67$^{hi}$) | Skin CD69$^-$ | 2.0 (1.7, 2.3) |
| Percent production from self-renewal | | CD4$^+$ EM | Quiescent (Ki67$^{low}$) | LN CD4$^+$ EM | 26 (22, 30) |
| | LP | CD4$^+$CD69$^+$ T$_{RM}$ | Neutral (Ki67$^{int}$) | LN CD4$^+$ EM | 28 (22, 34) |
| | | CD4$^+$ EM | Division-linked (Ki67$^{hi}$) | LN CD4$^+$ EM | 55 (50, 62) |
| | Skin | CD4$^+$CD69$^+$ T$_{RM}$ | Division-linked (Ki67$^{hi}$) | Skin CD69$^-$ | 52 (46, 59) |
| Ki67 lifespan (days) | | CD4$^+$ EM | Quiescent (Ki67$^{low}$) | LN CD4$^+$ EM | 2.6 (2.3, 2.9) |
| | LP | CD4$^+$CD69$^+$ T$_{RM}$ | Neutral (Ki67$^{int}$) | LN CD4$^+$ EM | 2.0 (1.7, 2.3) |
| | | CD4$^+$ EM | Division-linked (Ki67$^{hi}$) | LN CD4$^+$ EM | 2.8 (2.5, 3.2) |
| | Skin | CD4$^+$CD69$^+$ T$_{RM}$ | Division-linked (Ki67$^{hi}$) | Skin CD69$^-$ | 2.7 (2.4, 3.0) |

## Kinetics of mTom expression derived from Cd4-FR mice

We used a simple homogeneous ODE model to describe the kinetics of mTom$^+$ and mTom$^-$ cells by tracking their loss (per capita $\delta$), self-renewal (per capita rate $\rho$), and supplementation from a precursor population at constant total rate $\theta$ and with label content $f_{mTom}$, described empirically (see section 'Model fitting' below):

$$\frac{d}{dt}mTom^+ = \theta \cdot f_{mTom}(t) - (\delta - \rho) \cdot mTom^+$$
$$\frac{d}{dt}mTom^- = \theta \cdot (1 - f_{mTom}(t)) - (\delta - \rho) \cdot mTom^-$$

## Modelling the label trajectories derived from Ki67-DIVN mice

The model we used to describe data derived from the Ki67-DIVN mice was is similar to that above, with the same parameters $\theta$, $\delta$, and $\rho$, but included the kinetics of Ki67 expression (see equations below). Following mitosis, Ki67 protein levels continuously decrease. However, in standard flow cytometry analyses, cells are classified into two categories: Ki67$^{high}$ and Ki67$^{low}$. To model the transition between these states, we used the linear chain technique (*MacDonald, 1978*) and introduced intermediate

Ki67$^{high}$ compartments as in previous studies (**Bullock et al., 2024**), allowing us to characterise the residence time within the Ki67$^{high}$ compartment as gamma-distributed with low variance, with 12 intermediate Ki67 levels ($\ell$). The rate at which Ki67$^{high}$ cells move between these intermediate stages is given by $\ell\beta$, such that the expected time that a cell spends within the Ki67$^{high}$ gate before transitioning to the Ki67$^{low}$ gate is $1/\beta$. For each precursor/target pair, we examined three influx modes: one where new immigrant cells are Ki67$^{low}$ (quiescent), another where their Ki67 expression mirrors the precursor's, and a third where they are fully Ki67$^{high}$. To represent these modes we used a parameter ($k_f$) which was set to either zero, the mean Ki67$^{high}$ fraction within the precursor, or 1, respectively. In the model, the subscripts denote the Ki67 state or expression level (0 is Ki67$^{low}$; 1 ...$\ell$ are Ki67$^{high}$):

$$\frac{d}{dt}\text{YFP}_\ell^+ = \theta \cdot f_{\text{YFP}}(t) \cdot k_f - (\delta + \ell\beta + \rho) \cdot \text{YFP}_\ell^+ + 2\rho\sum_{j=0}^{\ell}\text{YFP}_j^+$$
$$\frac{d}{dt}\text{YFP}_\ell^- = \theta \cdot (1 - f_{\text{YFP}}(t)) \cdot k_f - (\delta + \ell\beta + \rho) \cdot \text{YFP}_\ell^+ + 2\rho\sum_{j=0}^{\ell}\text{YFP}_j^-$$
$$\frac{d}{dt}\text{YFP}_j^i = \ell\beta \cdot \text{YFP}_{j+1}^i - (\delta + \rho + \ell\beta) \cdot \text{YFP}_j^i \qquad \text{for } i \in \{+,-\} \text{ and } 1 \leq j < \ell$$
$$\frac{d}{dt}\text{YFP}_0^+ = \theta \cdot f_{\text{YFP}}(t) \cdot (1 - k_f) + \ell\beta \cdot \text{YFP}_1^+ - (\delta + \rho) \cdot \text{YFP}_0^+$$
$$\frac{d}{dt}\text{YFP}_0^- = \theta \cdot (1 - f_{\text{YFP}}(t)) \cdot (1 - k_f) + \ell\beta \cdot \text{YFP}_1^- - (\delta + \rho) \cdot \text{YFP}_0^-$$

## Model fitting

### Empirical functions describe label dynamics in precursor populations

We described the trajectories of YFP and mTom expression within the candidate precursors with simple functions based on exponentials. In most populations, the labelled fractions ($f_{\text{mTom}}$ and $f_{\text{YFP}}$), both decreased over time, each described with a curve of the form

$$f(t) = ae^{-bt}. \tag{1}$$

We used a saturating exponential increase function for the YFP$^+$ fraction within LN-derived naive CD4 T cells,

$$f(t) = a(1 - e^{-bt}). \tag{2}$$

For each precursor, we fitted these functions to its data alone using FME in *R* and in Python using the curvefit() function from the package SciPy (**Virtanen et al., 2020**). These fits are shown in **Figure 2D**. The best-fit parameters (see table below) and associated uncertainties were used to inform the priors for fitting precursor and target trajectories simultaneously using our Bayesian approach.

| Population | $a_y$ | $b_y$ | $a_t$ | $b_t$ | $k_f$ |
|---|---|---|---|---|---|
| LN CD4 CM | 0.286 | 0.001 | 0.857 | 0.011 | 0.2 |
| LN CD4 EM | 0.289 | 0.002 | 0.833 | 0.007 | 0.12 |
| LN CD4 Naive | 0.372 | 0.081 | 0.919 | 0.030 | 0.16 |
| SK CD69$^-$ | 0.258 | 0.001 | 0.542 | 0.007 | 0.21 |
| LP CD69$^-$ | 0.449 | 0.007 | 0.457 | 0.007 | 0.0047 |

### Initialising the system before describing post-treatment kinetics of labelled populations

Motivated by observations (**Figure 2A**), we assumed all target populations in skin and LP were at homeostatic equilibrium, and with constant levels of Ki67. To ensure the system was at this equilibrium before tamoxifen treatment, for each set of parameter values we used the steady-state assumption and observed Ki67 fraction (estimated by taking the mean of the observed values in each population) to eliminate two parameters from the model (see below). We then utilised the Runge–Kutta 45 ODE solver within Stan, setting all species to zero, and running for $10^4$ days to drive the system to this prescribed steady state. The labeled fractions of cells in the target population at d5 post treatment were free parameters $\epsilon_{\text{YFP}}$ and $\epsilon_{\text{mTom}}$, which reflected the cumulative effect of

label induction up to that point. Cells were then distributed into labelled and unlabelled populations using these parameters.

## Modelling noise and mouse-to-mouse variation

In addition to the empirically described trajectories of label content within each precursor population, from the model we derived the four proportions of interest for the target: the mTom$^+$ fraction, the YFP$^+$ fraction, and the Ki67$^{high}$ fractions within YFP$^{+/-}$ cells. As all of these observations lay within the interval (0, 1), we logit-transformed them to normalise residuals. The normally-distributed noise within each dataset was defined with six separate standard deviations, $\sigma$, each themselves described with normal priors. Fits were obtained using all six time series simultaneously (summing the log likelihoods). For performing statistical comparisons of different precursor–target relationships using LOO-IC, we used only the log-likelihoods for the four quantities derived from the target cell kinetics; this ensured we were comparing descriptions of the same observations.

## Prior distributions of parameters

By assuming that the population was at steady state and that the Ki67 fraction was constant, we could eliminate two of the five parameters ($\theta$, $\beta$, $\rho$, $\delta$, and $k_F$, the fraction of immigrant cells that are assumed to enter as Ki67$^{high}$).

All parameters were sampled from a range of values such that populations remained positive and finite. The prior on the Ki67 lifetime $1/\beta$ peaked at 3.25 days and constrained to remain between 2.5 and 4.5 days, based on previous studies. Based on previous studies (**Gossel et al., 2017**; **Bullock et al., 2024**), the influx rate $\theta$ was constrained to be between 0.05% and 10% of the total pool size. We set broad priors on the initialisation induction efficiencies of YFP and mTom ($\epsilon_{YFP}$ and $\epsilon_{mTom}$), with upper limits of 1.

All code and data can be obtained at https://github.com/elisebullock/CD4TRM.

## Estimating average cell residence times using Ki67

Consider a cell population at steady-state numbers $N$ with cells dividing at an average *per capita* rate $\rho$, leaving the population through death, egress, or differentiation at average *per capita* rate $\delta$, and being replenished from a precursor population at total rate $\theta$:

$$\frac{dN}{dt} = \theta + (\rho - \delta)N = 0 \implies \rho N + \theta = \delta N. \tag{3}$$

Assume a fraction $f$ of immigrants enter the population as Ki67$^{high}$. We know that Ki67 is expressed for a time $T \simeq 3 - 4$ days during and after division. Then the number of cells within the population as a whole that are Ki67$^{high}$, K$^+$, is constant and equal to the number that were produced by division or entered as Ki67$^{high}$ in the last $T$ days, and survived to the present:

$$K^+ = \int_0^T (2\rho N + f\theta)\, e^{-(\delta+\rho)(T-s)} ds = \frac{2\rho N + f\theta}{\delta + \rho}(1 - e^{-(\delta+\rho)T}), \tag{4}$$

The Ki67$^{high}$ proportion observed within the total population at any time is therefore also constant and equal to $k = K^+/N$

$$k = \frac{2\rho + f\theta/N}{\delta + \rho}(1 - e^{-(\delta+\rho)T}). \tag{5}$$

In the general case where influx is substantial, we need an independent estimate of the daily proportional replacement through influx, $\theta/N$, which by **Equation 3** is equal to the daily net loss rate $\lambda = \delta - \rho$. Then, eliminating the division rate $\rho$,

$$k = \frac{2(\delta - \lambda) + f\lambda}{2\delta - \lambda}(1 - e^{-(2\delta-\lambda)T}). \tag{6}$$

Given an observed value of $k$, the known value of $T$, an estimate of the replacement (influx) rate $\lambda$, and a value for $f$ (reflecting the extent to which incoming cells are recently divided), **Equation 6** can be solved numerically for the cell lifespan $\tau = 1/\delta$.

We can consider some simpler limiting cases. If most cell production occurs through self renewal rather than influx, then $\rho \simeq \delta$ and $\lambda \simeq 0$. **Equation 6** then becomes $k \simeq (1 - e^{-2T/\tau})$. This yields a simple formula that relates the Ki67 fraction within a closed or nearly closed population to the mean lifespan of its constituent cells:

$$\tau = \frac{-2T}{\ln(1-k)} \simeq \frac{2T}{k}, \tag{7}$$

where the approximation holds if $k$ is small. If influx cannot be disregarded, but we have an indication that cell lifespans are much longer than the duration of Ki67 expression, $\tau \gg T$ (which will be likely if $k$ is small, suggesting slow turnover), then $e^{-(2\delta-\lambda)T} \simeq 1 - T(2\delta - \lambda)$ and **Equation 6** gives

$$k \simeq T(\lambda(f - 2) + 2/\tau) \implies \tau \simeq \frac{2T}{k + \lambda T(2 - f)}. \tag{8}$$

Rough bounds on the mean cell lifespan are then

$$\frac{2T}{k + 2\lambda T} < \tau < \frac{2T}{k + \lambda T}, \tag{9}$$

spanning the cases $f = 0$ (all newly recruited cells are quiescent) and $f = 1$ (all new immigrants are recently divided; for example, having recently been antigen stimulated). A value of $f$ between 0 and 1 might also arise if the mean duration of Ki67 expression on new immigrants is shorter than the 3–4 days expected on very recently-divided cell, reflecting the possibility that they last divided as precursors less than $T$ days before entering the population.

