## [Editor Report · eLife Assessment]

This article provides a **compelling** and rigorous quantitative analysis of the turnover and maintenance of CD4^+^ tissue-resident memory T cell clones, in the skin and the lamina propria. It provides a **fundamental** advance in our understanding of CD4 T cell regulation. Interestingly, in both tissues, maintenance involves an influx from progenitors on the time scale of months. The evidence that is based on fate mapping and mathematical inference is strong, although open questions on the interpretation of the Ki67-based fate mapping remain.

---

## [Referee Report · Reviewer #1 (Public review)]

Summary:

Compelling and clearly described work that combines two elegant cell fate reporter strains with mathematical modelling to describe the kinetics of CD4+ TRM in mice. The aim is to investigate the cell dynamics underlying maintenance of CD4+TRM.

The main conclusions are that (1) CD4+ TRM are not intrinsically long-lived (2) even clonal half lives are short: 1 month for TRM in skin, even shorter (12 days) for TRM in lamina propria (3) TRM are maintained by self-renewal and circulating precursors.

Strengths:

(1) Very clearly and succinctly written. Though in some places too succinctly! See suggestions below for areas I think could benefit from more detail.

(2) Powerful combination of mouse strains and modelling to address questions that are hard to answer with other approaches.

(3) The modelling of different modes of recruitment (quiescent, neutral, division linked) is extremely interesting and often neglected (for simpler neutral recruitment).

Comments on revised version: This reviewer is satisfied with the author responses and the changes made in the manuscript.

---

## [Referee Report · Reviewer #2 (Public review)]

This manuscript addresses a fundamental problem of immunology - the persistence mechanisms of tissue-resident memory T cells (TRMs). It introduces a novel quantitative methodology, combining the in vivo tracing of T cell cohorts with rigorous mathematical modeling and inference. Interestingly, the authors show that immigration plays a key role for maintaining CD4+ TRM populations in both skin and lamina propria (LP), with LP TRMs being more dependent on immigration than skin TRMs. This is an original and potentially impactful manuscript.

Comments on revised version: This reviewer is satisfied with the author responses and the changes made in the manuscript.

---

## [Author Response]

The following is the authors’ response to the original reviews

**Reviewer #1 (Public review):**
Summary:Compelling and clearly described work that combines two elegant cell fate reporter strains with mathematical modelling to describe the kinetics of CD4+ TRM in mice. The aim is to investigate the cell dynamics underlying the maintenance of CD4+TRM.The main conclusions are that:(1) CD4+ TRM are not intrinsically long-lived.(2) Even clonal half-lives are short: 1 month for TRM in skin, and even shorter (12 days) for TRM in lamina propria.(3) TRM are maintained by self-renewal and circulating precursors.Strengths:(1) Very clearly and succinctly written. Though in some places too succinctly! See suggestions below for areas I think could benefit from more detail.(2) Powerful combination of mouse strains and modelling to address questions that are hard to answer with other approaches.(3) The modelling of different modes of recruitment (quiescent, neutral, division linked) is extremely interesting and often neglected (for simpler neutral recruitment).Weaknesses/scope for improvement:(1) The authors use the same data set that they later fit for generating their priors. This double use of the same dataset always makes me a bit squeamish as I worry it could lead to an underestimate of errors on the parameters. Could the authors show plots of their priors and posteriors to check that the priors are not overly-influential? Also, how do differences in priors ultimately influence the degree of support a model gets (if at all)? Could differences in priors lead to one model gaining more support than another?

We now show the priors and posteriors overlaid in Figure S2. The posteriors lie well within the priors, giving us confidence that the priors are not overly influential.

(2) The authors state (line 81) that cells were "identified as tissue-localised by virtue of their protection from short-term in vivo labelling (Methods; Fig. S1B)". I would like to see more information on this. How short is short term? How long after labelling do cells need to remain unlabelled in order to be designated tissue-localised (presumably label will get to tissue pretty quickly -within hours?). Can the authors provide citations to defend the assumption that all label-negative cells are tissue-localised (no false negatives)?And conversely that no label-positive cells can be found in the tissue (no false positives)? I couldn't actually find the relevant section in the methods and Figure S1B didn't contain this information.

We did describe the in vivo labeling in the first section of Methods (it was for 3 mins before sacrifice). The two aims of Fig S1B were to show the gating strategy (label-positive and negatives from tissue samples were clearly separated) and to address the false-positive issue. Less than 3% of cells in our tissue samples were positive; therefore, at most 3% of truly tissue-resident cells acquired the i.v. label, and likely less. Excluding those (as we did) therefore makes little difference to our analyses in terms of cell numbers. False negative rates are expected to be extremely low; labeling within circulating cells is typically >99% (see refs in Methods).

(3) Are the target and precursor populations from the same mice? If so is there any way to reflect the between-individual variation in the precursor population (not captured by the simple empirical fit)? I am thinking particularly of the skin and LP CD4+CD69- populations where the fraction of cells that are mTOM+ (and to a lesser extent YFP+) spans virtually the whole range. Would it be nice to capture this information in downstream predictions if possible?

This is a great point. We do indeed isolate all populations from each mouse. We are very aware of the advantages of using this grouping of information to reduce within-mouse uncertainty – we employ this as often as we can. The issue here was that the label content within the tissue (target) at any time depends on the entire trajectory of the label frequency in the precursor, in that mouse, up to that point. We can’t identify this curve for each animal individually – so we are obliged to use a population average.

To mitigate this lack of pairing we do take a very conservative approach and fit this empirical function describing the trajectories of YFP and mTom in precursors at the same time as the label kinetics in the target; that is, we account for uncertainty in label influx in our fits and parameter estimates.

Another issue is that to be sure that we are performing model selection appropriately, we only use the distribution of the likelihood on the target observations when comparing support for different precursors with LOO-IC. If we had been able to pair the precursor and target data in some way, the two would then be entangled and model comparison across precursors would not be possible.

We’ve added some of this to the discussion.

(4) In Figure 3, estimates of kinetics for cells in LP appear to be more dependent on the input model (quiescent/neutral/division-linked) than the same parameters in the skin. Can the authors explain intuitively why this is the case?

This is a nice observation and it has a fairly straightforward explanation. As we pointed out in the paper, estimated rates of self renewal become more sensitive to the mode of recruitment the greater the rate of influx. If immigrants are quiescent, all Ki67 in the tissue has to be explained by self renewal. If all new immigrants are Ki67 high, the estimate of the rate of self renewal within the tissue will be lower. Across the board, the estimated rates of influx into gut were consistently higher than those in skin, and so the sensitivity of parameters to the mode of recruitment was much more obvious at that site.

The importance of this trade-off for the division linked model can also be seen when you look at the neutral and quiescent models; they give similar parameter estimates because the Ki67 levels within all precursor populations were all less than 25% and so those two modes of recruitment are difficult to distinguish.

(5) Can the authors include plots of the model fits to data associated with the different strengths of support shown in Figure 4? That is, I would like to know what a difference in the strength of say 0.43 compared with 0.3 looks like in "real terms". I feel strongly that this is important. Are all the fits fantastic, and some marginally better than others? Are they all dreadful and some are just less dreadful? Or are there meaningful differences?

This is another good point (and from the author recommendations list, is your most important concern).

We find that a fairly common issue is that models that are clearly distinguished by information criteria or LRTs can often give visually quite similar fits. Our experience is that this is partly due to the fact that models are usually fit on transformed scales (e.g. log for cell counts, logit for fractions) to normalise residuals, and this uncertainty is compressed when one looks at fits on the observed scale (e.g. linear). Another issue in our case is that for each model (precursor, target, and mode of recruitment) we fit 6 time courses simultaneously. Visual comparisons of fits of different models can then be a little difficult or misleading; apparently small differences in each fitted timecourse can add up to quite significant changes in the combined likelihood. We added this to the Discussion.

The number of models is combinatorial (Fig. 4) so showing them all seems a bit cumbersome. But now in the supporting information (Fig. S3), for each target we show the best, second best, and the worst model fits overlaid, to give a sense of the dynamic range of the models we considered. As you will now see, visual differences among the most strongly supported models were not huge (but refer to our point just above). Measures of out-of-sample prediction error (LOO-IC) discriminated between these models reasonably well, though (weights shown in Fig. 4).

It’s also worth mentioning here that we have substantially greater confidence in the identity of the precursors than in the precise modes of recruitment - you can see this clearly in the groupings of weights in Figure 4A. We did comment on this in the text but now emphasise it more.

(6) Figure 4 left me unclear about exactly which combinations of precursors and targets were considered. Figure 3 implies there are 5 precursors but in Figure 4A at most 4 are considered. Also, Figure 4B suggests skin CD69- were considered a target. This doesn't seem to be specified anywhere.

Thanks for pointing this out. When we were considering CD4+ EM in bulk as target, this population includes CD69- cells; in those fits, therefore, we couldn't use CD69- as a precursor. We now clarify this in the caption. Thanks also for the observation about Figure 4B; we didn’t consider CD69- cells as a target, so we’ve also made that clearer.

**Reviewer #2 (Public review):**
This manuscript addresses a fundamental problem of immunology - the persistence mechanisms of tissue-resident memory T cells (TRMs). It introduces a novel quantitative methodology, combining the in vivo tracing of T-cell cohorts with rigorous mathematical modeling and inference. Interestingly, the authors show that immigration plays a key role in maintaining CD4+ TRM populations in both skin and lamina propria (LP), with LP TRMs being more dependent on immigration than skin TRMs. This is an original and potentially impactful manuscript. However, several aspects were not clear and would benefit from being explained better or worked out in more detail.(1) The key observations are as follows:a) When heritably labeling cells due to CD4 expression, CD4+ TRM labeling frequency declines with time. This implies that CD4+ TRMs are ultimately replenished from a source not labeled, hence not expressing CD4. Most likely, this would be DN thymocytes.

That’s correct.

b) After labeling by Ki67 expression, labeled CD4+ TRMs also decline - This is what Figure 1B suggests. Hence they would be replaced by a source that was not in the cell cycle at the time of labeling. However, is this really borne out by the experimental data (Figure 2C, middle row)? Please clarify.(2) For potential source populations (Figure 2D): Please discuss these data critically. For example, CD4+ CD69- cells in skin and LP start with a much lower initial labeling frequency than the respective TRM populations. Could the former then be precursors of the latter?A similar question applies to LN YFP+ cells. Moreover, is the increase in YFP labeling in naïve T cells a result of their production from proliferative thymocytes? How well does the quantitative interpretation of YFP labeling kinetics in a target population work when populations upstream show opposite trends (e.g., naïve T cells increasing in YFP+ frequency but memory cells in effect decreasing, as, at the time of labeling, non-activated = non-proliferative T cells (and hence YFP-) might later become activated and contribute to memory)?

These are good (and related) points. We've added some text to the discussion, paragraphs 2 and 3; we reproduce it here, slightly expanded.

Fig 1B was a schematic but did faithfully reflect the impact of any waning of YFP in precursor on its kinetic in the targets. However, in our experiments, as you noted, the kinetics of YFP in most of the precursor populations were quite flat. This was due in part to memory subsets being sustained by the increasing levels of YFP within naïve cells from the cohort of thymocytes labeled during treatment. There is also likely some residual permanent labeling of lymphocyte progenitor populations. We discussed this in Lukas Front Imm 2023. (The latter is not a problem; all that matters for our analysis is that we generate a reasonable empirical description of the label kinetics in naive cells, however it arises). YFP is therefore not cleanly washed out in the periphery; and so for models with circulating memory as the tissue precursor, the flatness of their YFP curves leads to rather flat curves in the tissues.

The mTom labelling was more informative as it was clearly diluted out of all peripheral populations by mTom-negative descendants of thymically-derived cells, as you point out in (a).

Regarding (2), re: interpreting the initial levels of labels in precursors and targets. The important point here is that YFP and mTom were induced quickly in all populations we studied; therefore our inferences regarding precursors and targets aren’t informed by the initial levels of levels in each. (Imagine a slow precursor feeding a rapidly dividing target; YFP levels in the former would start lower than those in the latter). The causal issue that we think you’re referring to would matter if one expects the targets to begin with no label at all; for instance, in our busulfan chimeric mouse model (e.g. Hogan PNAS 2015) new, thymically derived ‘labelled’ (donor) cells progressively infiltrate replete ‘unlabelled’ (host) populations. In that case, one can immediately reject certain differentiation pathways by looking the sequence of accrual of donor cells in different subsets.

The trends in YFP and mTom frequencies after treatment do matter for pathway inference, though, because precursor kinetics must leave an imprint on the target. For the case you mentioned, with opposite trends in label kinetics, such models would unlikely to be supported strongly; indeed, we never saw strong support for naïve cells (strongly increasing YFP) as a direct precursor of TRM (fairly flat).

We’ve added a condensed version of this to the Discussion.

(3) Please add a measure of variation (e.g., suitable credible intervals) to the "best fits" (solid lines in Figure 2).

Added.

(4) Could the authors better explain the motivation for basing their model comparisons on the Leave-OneOut (LOO) cross-validation method? Why not use Bayesian evidence instead?

Bayes factors are very sensitive to priors and are either computationally unstable if calculated with importance sampling methods, or very expensive to calculate, if ones uses the more stable bridge sampling method. (We also note that fitting just a single model here takes a substantial amount of time). Further, using BF can be unreliable unless one of the models is close to the 'true' data generating model; though they seem to work well, we can be sure that none of our models are! For us, a more tractable and real-world selection criterion is based on the usefulness of a model, for which predictive performance is a reasonable proxy. In this case the mean out-of-sample prediction error (which LOO-IC reflects) is a wellestablished and valid means of ascribing support to different models.